# Active expiration reduces hypercapnia in lung failure – results of the prospective interventional ActiveEx study and development of a prototype device for automated application

Denis Witham[1,☯,¤], Julius Valentin Kunz[1,☯], Ann-Christin Krebs[2], Karoline Brückner[1], Roland Körner[1], Mareen Pigorsch[1,3], Kai-Uwe Eckardt[1], Paul Uwe Thamsen[2], Philipp Enghard[1,‡,*], Julija Peter[2,‡]

1 Charité – Universitätsmedizin Berlin, corporate member of Freie Universität Berlin, Humboldt Universität zu Berlin, and Berlin Institute of Health; Department of Nephrology and Medical Intensive Care, 10117 Berlin, Germany, 2 Technical University of Berlin – Institute of Fluid Mechanics and Technical Acoustics of the Technical University of Berlin, Berlin, Germany, 3 Charité – Universitätsmedizin Berlin, corporate member of Freie Universität Berlin, Humboldt Universität zu Berlin, Institute of Biometry and Clinical Epidemiology, 10117 Berlin, Germany

☯ These authors contributed equally to this work.
‡ These authors also contributed equally to this work.
¤ Current address: Vivantes Klinikum am Urban, Department of Cardiology, Diabetology, and Medical Intensive Care, Berlin, Germany
* Philipp.Enghard@charite.de

## Abstract

### Background

This study investigates the efficacy of Intermittent Abdominal Pressure Ventilation (IAPV) and Expiratory Rib Cage Compression (ERCC) in reducing hypercapnia among critically ill, mechanically ventilated patients. It also assesses the feasibility of automating these techniques using a self-developed, ventilator-synchronized device on a dummy.

### Methods

This single-arm feasibility study was conducted in two intensive care units at Charité – Universitätsmedizin Berlin, including critically ill patients with hypercapnic lung failure, with additional lab testing at the Technical University of Berlin. Manual IAPV and ERCC were applied to patients, and automation feasibility was tested on a dummy using prototype device. Primary outcomes included changes in tidal volume and partial pressure of carbon dioxide ($PaCO_2$) and device effectiveness at different PEEP levels.

**Data availability statement:** All relevant data are within the manuscript and its Supporting information files.

**Funding:** The author(s) received no specific funding for this work.

**Competing interests:** The authors have declared that no competing interests exist.

## Results

In nine hypercapnic patients, manual IAPV increased tidal volume from 5.72 to 8.85 mL/ kg predicted bodyweight (p < 0.001, relative effect (CI) 0.93 (0.79–1.06)) and ERCC from 5.79 to 8.13 mL/ kg predicted bodyweight (p < 0.001, relative effect (CI) 0.93 (0.80–1.05)). PaCO2 reduced after 20 minutes with both techniques (IAPV: from 65 to 52 mmHg, p < 0.01, relative effect (CI) 0.15 (0.01–0.28); ERCC: from 61 to 51 mmHg, p= < 0.01, relative effect (CI) 0.22 (0.07–0.37)). A transient decrease in oxygenation was fully and rapidly reversible. The automated device doubled tidal volumes in dummy simulations, with greater effectiveness at higher PEEP-levels.

## Conclusion

Manual and automated ventilator-synchronized IAPV and ERCC were associated with improved ventilation. Their potential role in managing hypercapnic respiratory failure—such as in weaning failure, obstructive lung disease, or neuromuscular weakness—remains a subject for future clinical research.
German Registry of Clinical Trials (DRKS00027397)

## Introduction

Respiratory failure can result from various etiologies such as acute respiratory distress syndrome (ARDS), chronic obstructive pulmonary disease (COPD), pneumonia, bronchial asthma, neurodegenerative disorders, and others. Inadequate ventilation leads to insufficient decarboxylation and respiratory acidosis, which, if severe, requires ventilatory support in an intensive care unit (ICU). Invasive ventilation in these patients itself can be challenging in severe bronchial obstruction or marked reduction of pulmonary compliance and resistance. This often forces clinicians to use high ventilatory driving pressures or apply extracorporal support for elimination of $CO_2$.

Two chest compression techniques, namely intermittent abdominal pressure ventilation (IAPV) and the more commonly applied expiratory rib cage compression (ERCC), also known as "squeezing" [1], may enhance ventilation. [2,3] IAPV involves applying a slight and short manual pressure to the medial upper abdomen during each expiratory cycle, resulting in an increased diaphragmatic upward motion. Similarly, ERCC utilizes manual pressure applied bilaterally to the ventrolateral ribcage during expiration.

To date, IAPV has been beneficially applied in patients with neurodegenerative disorders and respiratory muscle paresis [2,4], while ERCC has been studied more frequently in patients ventilated over an endotracheal tube due to various clinical conditions [1,5–9]. Explicit data on its effects on gas exchange and hemodynamics in patients with respiratory failure are rare and strongly depend on the study design [1].

In critical care IAPV and ERCC are typically administered by chest physiotherapists during short sessions per patient. Limited healthcare professional

resources pose a significant barrier to the standardized application of these techniques. Thus, there is potential for a device automating upper abdominal or ventrolateral thorax compression. Currently, "Luna Belt", equipped with a reversible filled air cushion performing IAPV, is available on the market, showing evidence of symptom improvement and enhanced ventilation in patients with neurodegenerative disorders primarily in home care settings. [2,4,10,11]

Here, we aimed at investigating whether the application of IAPV and ERCC can enhance ventilation and improve hypercapnia primarily in patients with ARDS. Furthermore, we constructed a prototype that allows automation of these techniques using a self-constructed device on a mannequin under simulated ICU conditions.

Given the lack of robust clinical data on IAPV and ERCC in acute hypercapnic respiratory failure, this study was intentionally designed as a pilot and feasibility trial. Its primary goal was to generate preliminary physiological insights, assess technical safety, and explore the potential for automated application in a controlled setting. It was not powered nor designed to evaluate long-term outcomes or to support confirmatory conclusions.

## Methods

The ActiveEx Study is a prospective, single-center, single-arm pilot study aiming at evaluating feasibility, safety, and clinical outcomes of IAPV and ERCC. The study took place across two ICUs within Charité – Universitätsmedizin Berlin. The recruitment period for this study started on December 12, 2021, and ended on June 22, 2022. The screening for eligible participants was conducted by the study team on the participating ICU wards using the hospital information system. Ethical approval for this study (No. EA2/241/21) was provided by the Ethical Committee of Charité – Universitätsmedizin Berlin, Germany on November 4, 2021. Registration in the German Registry of Clinical Trials (DRKS00027397) occurred on February 17, 2022. The trial adhered to the principles of the Declaration of Helsinki and written informed consent was obtained from patients or their legal representatives. No minors were included in this study.

To ensure patient confidentiality, exact ages were not reported; instead, all ages were categorized into decade ranges (e.g., 43 years → 40–50 years).

This study was designed as an exploratory, single-arm pilot trial with a planned enrollment of 20 patients. No formal sample size calculation was performed, consistent with the study's exploratory nature. During the course of the study, recruitment was stopped after 10 patients due to the consistent and pronounced treatment effects observed in this group. As no control group was included and further enrollment was not expected to substantially change the overall interpretation or add significant insights, the decision was made to discontinue recruitment. This decision was made before any formal analysis and without conducting any statistical hypothesis testing.

### Patient selection, procedures and outcomes

Inclusion criteria were aged 18 years or older, invasive mechanical ventilation with PEEP > 8 millibar (mbar), and hypercapnia (PaCO² ≥ 45 mmHg). Exclusion criteria included pneumothorax, thoracal fractures, traumatic injuries, indwelling thoracic or abdominal drains, extracorporeal lung replacement procedures, intra-abdominal infections, relevant wounds on thorax or abdomen, intra-abdominal/thoracic/gastrointestinal/intracerebral bleeding, hemophilia, severe thrombocytopenia (platelets <20/µl), acute liver failure (Bilirubin mg/dl > 8 and Quick < 50%), severe circulatory instability, and pregnancy. Investigators performed patient screening and enrolment in dialogue with the attending physician confirming eligibility.

All patients received standardized treatment based on current guideline recommendations. IAPV and ERCC were applied each for 20 minutes with a 20-minute interval in between by professionals of the study team, and data were collected before, during, and after these interventions (S1 Fig). Patients were ventilated with Draeger Evita V600. Ventilator settings, analgosedation, and norepinephrine dosing were maintained as defined by the treating physicians. The study's

primary endpoints were to assess the effect of IAPV and ERCC on tidal volumes and $p_aCO_2$. Furthermore, hemodynamics, peripheral oxygen saturation, Horowitz index ($PaO_2/FiO_2$), alveolo-arterial oxygen difference ($AADO_2$), intrinsic positive end-expiratory pressure (iPEEP), pulmonary compliance and resistance, and peak inspiratory pressure were documented immediately before, during, and after the intervention. Tracheobronchial secretions were collected and weighed before and immediately after IAPV and ERCC, using a standardized five-second suction period. The impact of IAPV and ERCC on diaphragmatic motion was assessed using standardized M-mode sonography. The ultrasound probe was placed in the anterior axillary line, and the diaphragm was continuously vertically in M-mode, hence documenting its movements in millimeters during ventilation. Diaphragmatic movement towards was measured before and while performing IAPV and ERCC. Manual compression pressure was estimated in mbar. The applied manual pressure was measured with air-filled, self-constructed cushions (S2 Fig). They were made out of silicone, with a diameter of about 86 mm. Two relative pressure sensors (1UNIK 5000 PTX 5072-TA-A2-CA-HO-PB, 0–300 mbar) were connected to the air-filled cushions. A microcontroller (Arduino UNO, 0.37 mbar resolution) was used for data acquisition. Invasive arterial and central venous pressure, heart rate, heart rhythm, and lactate levels served to evaluate hemodynamics. Two medical doctors and a nurse constituted the study team.

### Statistical analysis

Conducted as an investigator-initiated pilot study, this research aimed to assess the feasibility, safety, and effects of IAPV and ERCC, without sample size calculation due to its single-arm, exploratory design. Continuous variables were presented as median and IQR. To compare pre- and post-intervention measurements, consisting of metric paired data, we applied a nonparametric paired-sample t-test based on relative effects, that allows for heteroscedastic variances (R package nparcomp [12]). Outcome comparisons were based on the relative effect, with corresponding p-values reported for statistical inference without adjustment for multiple testing due to the exploratory character of the study. Values above or below 1/2 indicate that data during or after the intervention tend to be larger or smaller than before the intervention, respectively. R statistical software was used for analysis and graphical preparation of the data (The "R" Foundation for Statistical Computing, Vienna, Austria, Version R-4.2.2) [13].

### Results

Between December 12, 2021, and June 22, 2022, a total of 10 patients were enrolled in the study. Following the exclusion of one patient who withdrew consent (n = 1), the final analysis included nine patients (Fig 1). Recruitment was stopped after these 10 patients due to the consistent and marked effects observed, which were considered sufficient to meet the exploratory aims of the study.

Table 1 displays baseline characteristics of the patients. Eight patients were invasively ventilated due to Covid-19 associated ARDS (CARDS), and one due to exacerbated COPD. Eight patients were mechanically ventilated using mandatory PCV-BIPAP mode, and one was on CPAP with pressure support (Draeger V600 was used in all patients). All patients received analgosedation as defined by the treating physician and required low-dose catecholamine support.

### Increased decarboxylation by IAPV and ERCC

IAPV and ERCC (Fig 2) were manually performed by the study team for a duration of 20 minutes, each. The results indicate a notable improvement in the primary endpoint (ventilation and decarboxylation) both, for IAPV and for ERCC (Table 2). The interventions led to a substantial increase in tidal volumes. While performing IAPV, median tidal volumes expressed in mL per kg predicted body weight increased by 42%, i.e., from 6.23 (5.72, 6.86) mL/ kg predicted bodyweight to 8.85 (7.73, 10.54) mL/ kg predicted bodyweight (p=<0.001, relative effect (CI) 0.93 (0.79–1.06)) (Fig 3A). ERCC also

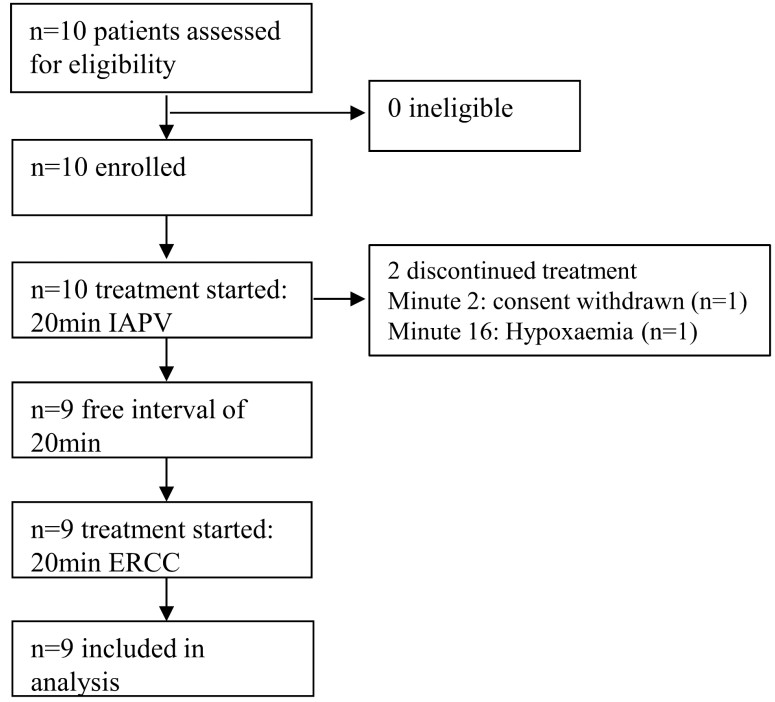

**Fig 1. Flowchart of the ActivEx-study.** After enrollment, IAPV was initiated in 10 patients. In one case, there was a consent withdrawn by the legal representative 2 minutes after the start. In another patient, the procedure had to be interrupted at the 16th minute due to moderate hypoxia, which was completely reversible in the subsequent 20-minute free interval, allowing for the administration of ERCC in this patient. In total, 9 patients were included in the evaluation.

led to an increase from 5.79 (5.40, 6.83) mL/ kg predicted bodyweight to 8.13 (7.27, 9.15) mL/ kg predicted bodyweight (p=<0.001, relative effect (CI) 0.93 (0.80–1.05)), equivalent to a change of 40,4% (Fig 3C).

Consequently, 20 minutes of IAPV demonstrated a considerable decrease in $p_aCO_2$ (Fig 3B) as median baseline dropped from 65 mmHg (54, 66) to 52 mmHg (39, 52) (p=<0.01, relative effect (CI) 0.15 (0.01–0.28)). Similarly, ERCC exhibited a substantial reduction in $p_aCO_2$, decreasing from 61 mmHg (52, 65) to 51 mmHg (40, 58) (p=<0.01, relative effect (CI) 0.22 (0.07–0.37)) after 20 minutes (Fig 3D). Respiratory acidosis decreased as IAPV led to an increase in pH from 7.35 (7.24, 7.38) to 7.43 (7.38, 7.48) and ERCC from 7.35 (7.28, 7.38) to 7.46 (7.38, 7.49). Tidal volumes of each patient before, during the intervention, and the percentage increase are depicted in S1 Table. The tidal volumes after the intervention were approximately the same as those before the intervention, as detailed in S2 Table.

The median manual compression pressure exerted by hand on the upper abdomen was 51 mbar (IQR: 41, 59) in IAPV and 84 mbar ventrolaterally on the ribcage (IQR: 71, 94) in ERCC. In normal palpation of the upper abdomen (i.e., to evaluate for acute pathologies in an emergency department), a comparable pressure to that one applied in IAPV, is used.

To further investigate the mode of action of IAPV and ERCC, diaphragmatic movement, tracheobronchial secretion, $AADO_2$, and intrinsic PEEP were assessed. Diaphragmatic motion increased from 1.10 cm (0.93, 1.18) to 1.65 cm (1.37, 2.04) (p=0.03, relative effect (CI) 0.76 (0.50–1.01)) during IAPV and from 1.14 cm (0.85, 1.38) to 1.34 cm (1.24, 1.46) (p=0.10, relative effect (CI) 0.71 (0.43–1.00)) during ERCC. We observed no effect on tracheobronchial secretions. Notably, seven out of eight patients exhibited no tracheobronchial secretions. Intrinsic PEEP remained constant. Following the IAPV intervention, $AADO_2$ increased from a median of 263 (IQR: 220–356) to 319 (IQR: 257–377) (p<0.01, relative effect (CI) 0.63 (0.57–0.69)), whereas after ERCC, it rose from 278 (IQR: 225–339) to 307 (IQR: 252–360) (p<0.001, relative

**Table 1. Baseline characteristics of the patients.**

| | Sex | Age | BMI | Days on invasive ventilation | Indication | Respirator settings | Pre-existing diseases |
|---|---|---|---|---|---|---|---|
| 1 | f | 50-60 | 46 | 7 | CARDS | BIPAP, 25/Min, FIO$_2$ 50%, PEEP 20mbar, dP 15mbar | AH, Follicular Lymphoma IIIA, Liver cirrhosis Child A |
| 2 | m | 50-60 | 22 | 42 | CARDS | BIPAP, 22/Min, FIO$_2$ 60%, PEEP 15mbar, dP 15mbar | DM |
| 3 | m | 60-70 | 31 | 20 | CARDS | BIPAP, 24/Min, FIO$_2$ 50%, PEEP 14mbar, dP 15mbar | Chronic Lymphatic Leukemia |
| 4 | m | 50-60 | 30 | 13 | CARDS | CPAP, 20/Min, FIO$_2$ 80%, PEEP 19mbar, dP 12mbar | PE, DVT |
| 5 | m | 60-70 | 35 | 22 | CARDS | BIPAP, 20/Min, FIO$_2$ 60%, PEEP 14mbar, dP 14mbar | OSAS, PE, DM |
| 6 | m | 70-80 | 28 | 11 | CARDS | BIPAP, 25/Min, FIO$_2$ 70%, PEEP 13mbar, dP 21mbar | Pulmonary fibrosis, AFIB, AH, DM, Hyperthyroidism |
| 7 | m | 80-90 | 29 | 38 | CARDS | BIPAP, 23/Min, FIO$_2$ 70%, PEEP 11mbar, dP 18mbar | unknown |
| 8 | m | 60-70 | 29 | 13 | CARDS | BIPAP, 18/Min, FIO$_2$ 80%, PEEP 13mbar, dP 9mbar | AMI, CABG, COPD, AH, Dyslipidemia |
| 9 | m | 60-70 | 23 | 8 | COPD | BIPAP, 9/MIN, FIO$_2$ 40%, PEEP 9mbar, dP 15mbar | COPD, AH |

BMI Body Mass Index, f female, CARDS Covid-19 Acute Respiratory Distress Syndrome, BIPAP Bi-level Positive Airway Pressure, FIO2 Inspiratory Oxygene Fraction, PEEP Positive Endexspiratory Pressure, mbar millibar, dP driving pressure, AH Arterial Hypertension, m male, DM Diabetes Mellitus, PE Pulmonary Embolism, DVT Deep Vene Thrombosis, OSAS Obstructive Sleep Apnea Syndrome, AFIB Atrial Fibrillation, AMI Acute Myocardial Infarction, CABG Coronary Artery Bypass Graft, COPD Chronic Obstructive Pulmonary Disease.

A)          B)

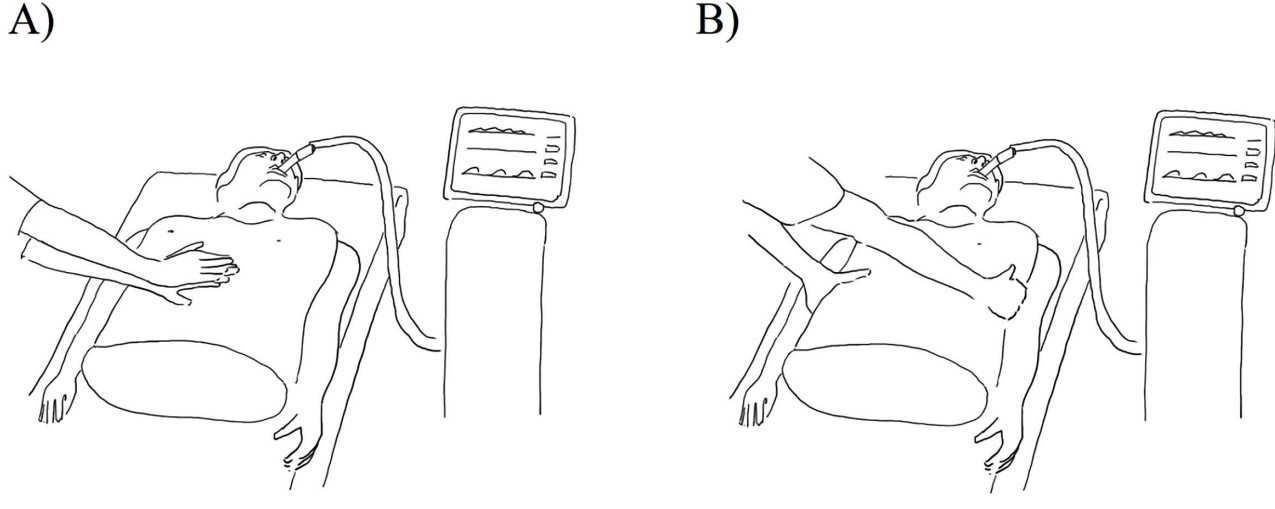

**Fig 2. Schematic drawing of IAPV (A) and ERCC (B).**

effect (CI) 0.61 (0.52–0.69)). Bioimpedance measurements in one patient indicated increased ventilation in dorsal areas where atelectasis and infiltration were seen in the patients CT, suggesting possible recruitment and redistribution of ventilation (S3 Fig). Other endpoints can be found in S2 Table.

**Table 2. Respiratory and hemodynamic outcome parameters.**

| | | IAPV N = 9 | | |
|---|---|---|---|---|
| | **Before[1]** | **During/ *Following*[1]** | **Relative effect[2] (CI)[3]** | **p-value[4]** |
| Tidal volume[5] [mL/ kg] | 6.23 (5.72, 6.86) | 8.85 (7.73, 10.54) | 0.93 (0.79–1.06) | <0.001 |
| pH value | 7.35 (7.24, 7.38) | *7.43 (7.38, 7.48)* | 0.80 (0.68–0.92) | <0.001 |
| $p_aCO_2$ [mmHg] | 65 (54, 66) | *52 (39, 52)* | 0.15 (0.01–0.28) | <0.01 |
| $P_aO_2$ [mmHg] | 82 (77, 92) | *64 (51, 81)* | 0.21 (−0.01–0.43) | 0.02 |
| $P_aO_2/FIO_2$ | 140 (134, 146) | *102 (80, 116)* | 0.12 (−0.13–0.38) | 0.02 |
| $AADO_2$ | 263 (220, 356) | *319 (257, 377)* | 0.63 (0.57–0.69) | <0.01 |
| Compliance [ml/cm $H_2O$] | 33 (23, 39) | 47 (33, 67) | 0.77 (0.68–0.86) | <0.001 |
| Diaphragmatic movement[6] [cm] | 1.10 (0.93, 1.18) | 1.65 (1.37, 2.04) | 0.76 (0.5–1.01) | 0.03 |
| Heartrate [bpm] | 86 (77, 97) | 83 (70, 97) | 0.44 (0.39–0.50) | 0.07 |
| MAP [mmHg] | 78 (76, 78) | 82 (80, 86) | 0.85 (0.63–1.07) | <0.01 |
| Manual applied pressure[7] [mbar] | | *51 (41, 59)* | | |
| | | **ERCC** N = 9 | | |
| Tidal volume[5] [mL/ kg] | 5.79 (5.40, 6.83) | 8.13 (7.27, 9.15) | 0.93 (0.80–1.05) | <0.001 |
| pH value | 7.35 (7.28, 7.38) | *7.46 (7.38, 7.49)* | 0.77 (0.56–0.95) | <0.01 |
| $p_aCO_2$ [mmHg] | 61 (52, 65) | *51 (40, 58)* | 0.22 (0.07–0.37) | <0.01 |
| $P_aO_2$ [mmHg] | 95 (77, 104) | *69 (63, 82)* | 0.19 (0.00–0.37) | 0.01 |
| $P_aO_2/FIO_2$ | 139 (131, 155) | *109 (102, 125)* | 0.16 (−0.08–0.41) | 0.01 |
| $AADO_2$ | 278 (225, 339) | *307 (252, 360)* | 0.61 (0.52 - 0.69) | <0.001 |
| Compliance [ml/cm $H_2O$] | 23 (23, 38) | 35 (26, 49) | 0.70 (0.62–0.79) | <0.001 |
| Diaphragmatic movement[6] [cm] | 1.14 (0.85, 1.38) | 1.34 (1.24, 1.46) | 0.71 (0.43–1.00) | 0.10 |
| Heartrate [bpm] | 87 (77, 100) | 82 (78, 99) | 0.71 (0.43–1.00) | 0.10 |
| MAP[mmHg] | 76 (76, 85) | 86 (82, 88) | 0.68 (0.46–0.90) | 0.11 |
| Manual applied pressure[7] [mbar] | | *84 (71, 94)* | | |

[1]Median (IQR);

[2]Nonparametric paired-sample t-test based on relative effects;

[3]Confidence interval;

[4]non-parametric paired-sample t-test; Tidal volume expressed in mL per kg predicted body weight;

[6]N = 8 (one missing value);

[7]N = 6 (3 missing values); $p_aCO_2$ = Partial Pressure of Carbon Dioxide in Arterial Blood; $P_aO_2$ = Partial Pressure of Oxygen in Arterial Blood, $FIO_2$ = Fraction of Inspired Oxygen, $AADO_2$ = Alveolar-arterial Oxygen Gradient, MAP = Mean Arterial Pressure

## Safety

Although IAPV and ERCC showed favorable effects respective to ventilation, all 8 CARDS patients experienced a decrease in oxygenation except for patient 9 with exacerbated COPD. In one case, IAPV was stopped before 20 minutes due to an ongoing decrease in $SPO_2$–80%. In the other cases with CARDS, $SPO_2$, $SaO_2$, and $paO_2$ only slightly declined regardless of the technique used. However, this deterioration was fully and immediately reversible within 20 minutes of follow-up in all cases. Blood gas analysis within three hours after the intervention, when considering the IQR range, indicate a modest improvement in oxygenation and decarboxylation: Arterial Oxygen Partial Pressure to Fraction of Inspired Oxygen ($PaO_2/FIO_2$) was (138 (133, 146) before vs. (141 (139, 172) afterwards (S2 Table)) and $p_aCO_2$ baseline was (59 mmHg (50, 62) vs. 63 mmHg (53, 66)). Hemodynamics showed a subtle trend towards increased blood pressure and reduced heart rate, as detailed in Table 2. No adverse effect of IAPV and ERCC on hemodynamics was recorded. Central venous pressure didn't change with relevance while performing IAPV and ERCC. In patient 9, small bowel ischemia

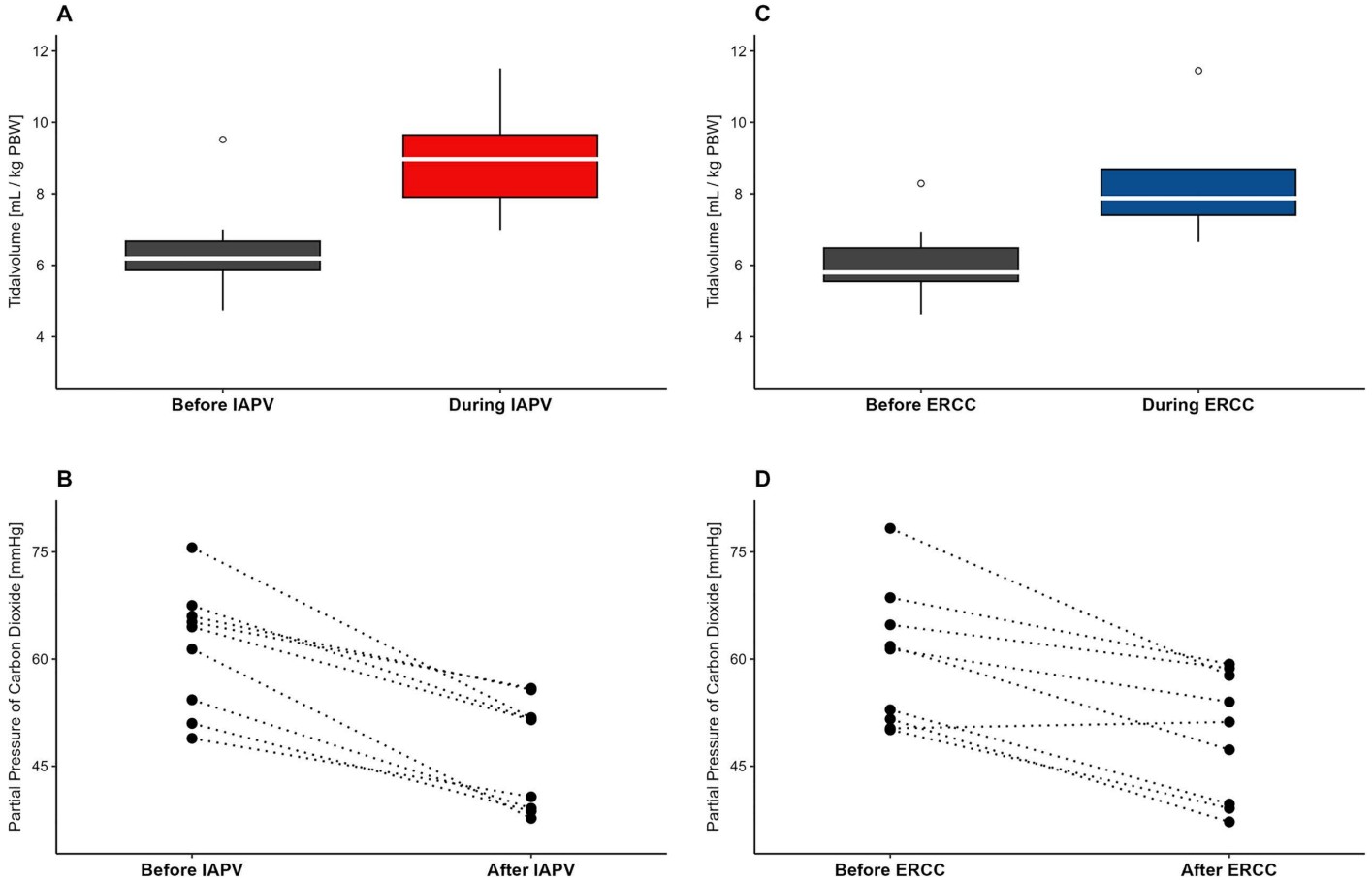

**Fig 3. Increased ventilation and decarboxylation by IAPV and ERCC. A)** Depicts median tidal volumes before and during IAPV. **B)** $p_aCO_2$ values of individual patients before and after IAPV **C)** Illustrates median tidal volumes before and during ERCC. **D)** $p_aCO_2$ values of individual patients before and after ERCC are shown.

occurred due to torsion without perforation on the 9th day after the study, which was treated surgically. Nonetheless, the patient was successfully transferred from the ICU 14 days after the study.

## Manual IAPV and ERCC in a simulated ICU environment and Evaluation of a prototype device

A "laboratory intensive care unit environment" was set up. So far, a physiognomic patient model (dummy) with implemented test lungs was intubated (tube diameter of 8.0 mm) and ventilated by a respirator (Draeger Savina 300). Ventilation mode was PCV-BIPAP (inspiratory time 1.3s, frequency 20/min, inspiratory termination of 25%).

During normal ventilation of the test lungs in PCV-BIPAP mode, tidal volumes increased when driving pressure was increased and decreased when PEEP was elevated under steady-state conditions without any intervention. This represents normally seen effects in clinical routine.

At a constant PEEP and driving pressure of 5 mbar, both ERCC and IAPV led to an increase of about 100% in tidal volume. Notably, when driving pressure was set constant at 5 mbar, but PEEP was increased stepwise during IAPV or ERCC, the beneficial rise in tidal volume induced by IAPV and ERCC got even higher, the higher PEEP was set.

A prototypic belt was constructed to automize IAPV and ERCC by reciprocal filling of implemented cushions during expiration (Fig 4A). Filling was synchronized to internal flow sensor signals extracted from the respirator. Volume flow to the cushions of the belt was provided by an external supply of compressed air. The prototype belt was fitted to the dummy. Length adjustments were realized by touch fasteners. Cushions could be positioned flexible at the belt to provide either IAPV (one cushion) or ERCC (two cushions). The cushions consisted of rubber and had a rectangular surface area of 110 mm x 120 mm (S2 Fig).

## Measurements, data traffic and control

To synchronize filling of cushions to respiratory cycles as described above, Draeger Savina 300 was connected to a PC using serial interface (Fig 4B) and the Medibus protocol. Pressure and filling were controlled by a pressure control valve (Festo VEAB-L-26-D12-Q4-V1-1R1). In addition, two magnetic valves (Bürkert 330-B-4,0-F-VA) were installed to

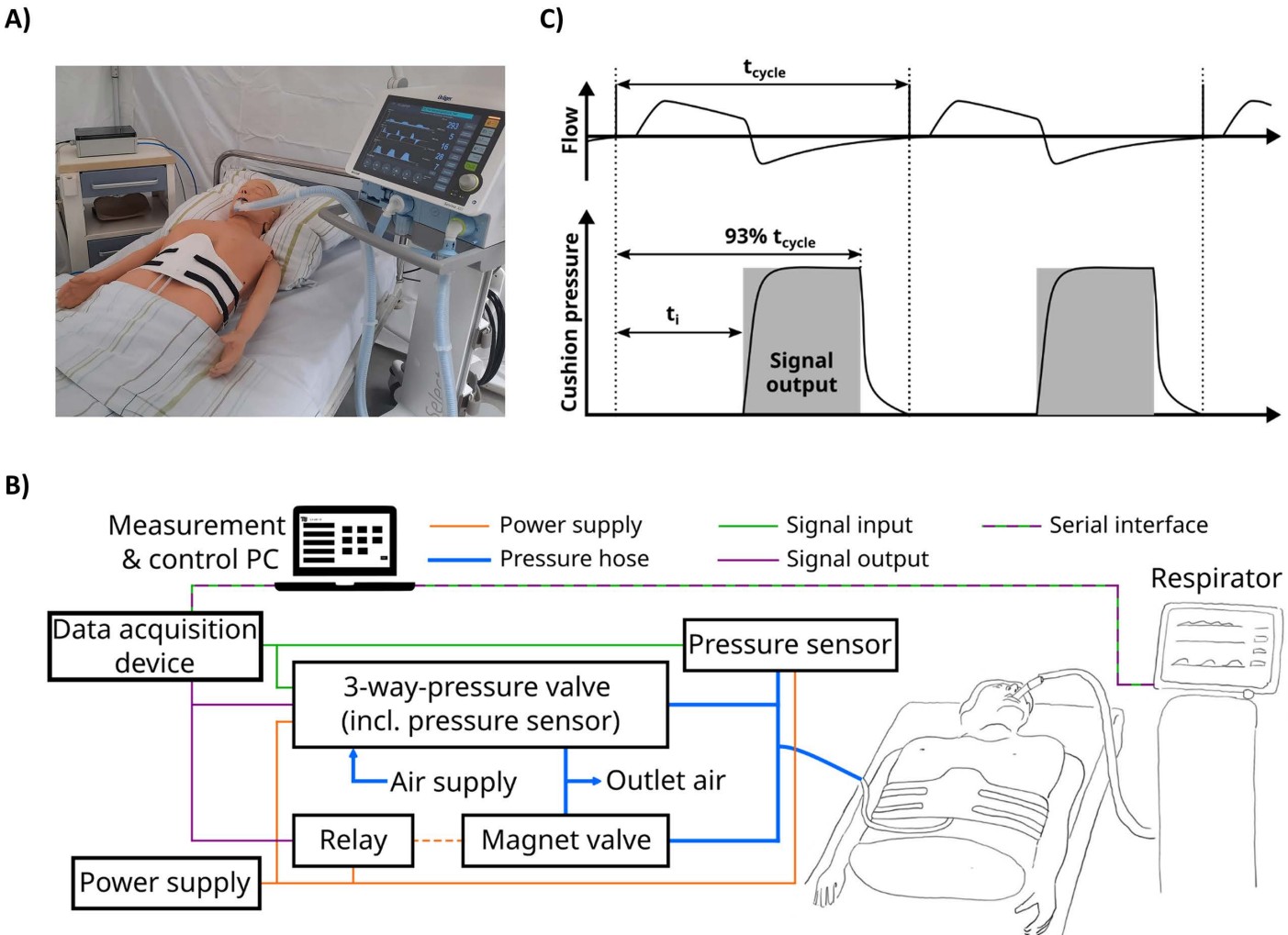

**Fig 4. Measurements, data traffic and laboratory intensive care unit environment: A)** Laboratory intensive care unit environment: a test bed with a patient model, our prototype belt and Draeger Savina 300. **B)** Scheme of sensors, measurement, and data transfer. **C)** Timing of filling and deflation depending on respiratory frequency and inspiratory time.

accelerate filling and deflation of the cushion. A second pressure sensor (Sensortechnics 218–08314) was used to check for measurements of the pressure control valve's pressure sensor.

Controlled process variable was tidal volume. The actuating variable was pressure in the air-filled cushions. The latter was limited to 160 mbar. Fig 4C outlines timing of filling and deflation dependent on respiratory frequency and inspiratory time. Ventilation of the test lungs is shown in the first curve ("flow"). Timing of cushion filling was synchronized to the respiratory cycle and cushion filling started with expiration. Active deflation began shortly before the end of expiration, i.e., at 93% of the respiratory cycle. In doing so, sensing of false inspiratory breathing effort could be prevented.

### Automated IAPV and ERCC

Tests with automated compressions executed by the constructed prototype belt showed that automated IAPV and ERCC are feasible and equivalent efficient compared to manual IAPV and ERCC. Also, automated IAPV and ERCC became constantly more effective with higher PEEP levels. Generated tidal volumes increased by 85% (automated IAPV, Fig 5) and 58% (automated ERCC, Fig 5) at a PEEP of 5mbar and driving pressure of 5mbar. When driving pressure remained constant at 5mbar, but PEEP was increased from 5mbar to 15mbar, automated IAPV led to increasing tidal volumes by 157% and ERCC led to an increase by 164% (Figs 5, S3 Table).

### Discussion

The ActiveEx Study demonstrates notable and comparable improvements in ventilation and reduction of hypercapnia by IAPV and ERCC, without increasing the peak inspiratory pressure or driving pressure. The mode of action seems to rely in some part on increased diaphragmatic motion. We hypothesize that the techniques essentially function as an artificial accessory respiratory muscle, working in synergy with the lung's elastic recoil forces.

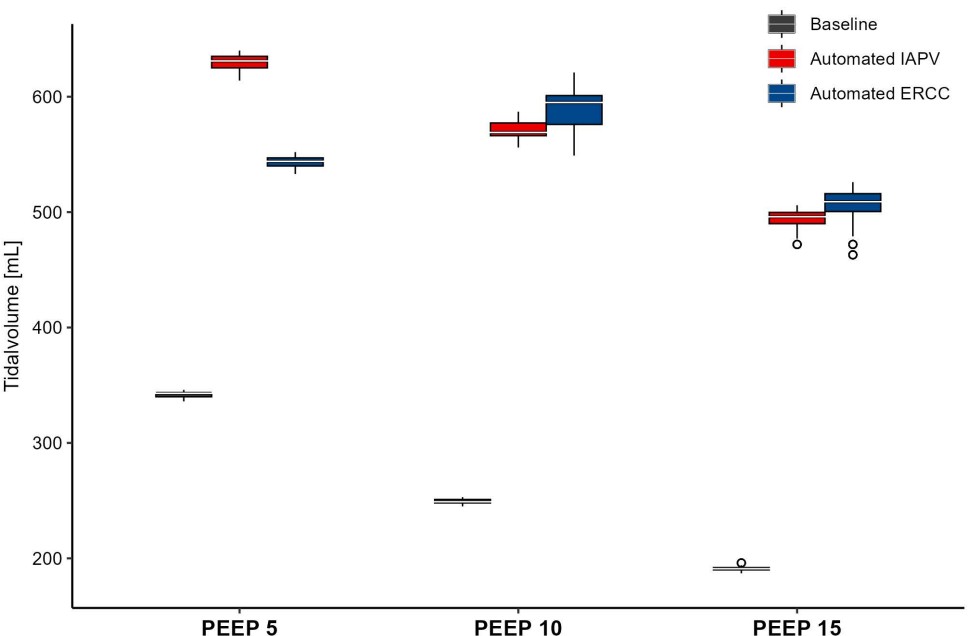

**Fig 5. Effect of automated chest compression on tidal volumes in a simulated intensive care environment: Comparison of tidal volumes over 120 mechanically delivered breaths at PEEP levels of 5, 10, and 15 mBar, without (Baseline) and with automated inspiratory assistance (IAPV and ERCC), under constant driving pressure (5 mBar).**

The study's findings suggest that IAPV and ERCC could serve as means to decrease inspiratory pressure, thereby potentially contributing to more lung-protective ventilation regimes in ARDS. This aspect might also be relevant for patients with obstructive respiratory diseases like COPD, where reducing ventilatory pressures can be challenging in order to maintain adequate $p_aCO_2$ levels [3].

Given preliminary indications of efficacy in enhancing $CO_2$ elimination, these techniques warrant further investigation regarding their potential to prevent intubation in patients with hypercapnic respiratory failure. Tolerance to such devices and synchronization to the patient's own ventilation are probably two issues. The applied pressures were approximately as high as those used during a physical palpation, suggesting that we can expect good tolerance even in awake patients. Synchronization to the patient's spontaneous breathing efforts would be crucial to ensure comfort and effectiveness. Our prototype device demonstrated the ability to synchronize well with the respiratory cycle during simulations. Future studies should investigate whether early application of IAPV and ERCC can effectively prevent intubation. Moreover, the potential of IAPV and ERCC to decrease hypercapnia holds promise for treating neurodegenerative diseases, where respiratory muscle weakness leads to decreased ventilation [4].

We observed a trend toward increased diaphragmatic movement during IAPV and ERCC. Since diaphragm dysfunction and atrophy frequently contribute to weaning failure in mechanically ventilated patients [14], such techniques may help engage the diaphragm. Whether this effect could support weaning processes remains a hypothesis to be tested in future studies.

The actual study found no evidence that IAPV or ERCC have any impact on clearance of tracheobronchial secretions as a potential reason for ameliorated ventilation.

It was the first study, directly comparing both techniques to each other, showing that both techniques are equally feasible. ERCC was equally effective, but on the price of higher compression pressures. This could promote IAPV as the potentially more suitable technique in critical care. Notably, the sequence of interventions was not randomized, with all patients receiving IAPV prior to ERCC. This fixed order may have introduced carryover or timing effects and limits the strength of direct comparative conclusions regarding clinical effectiveness.

The history of IAPV traces back to 1766, as detailed by Bach et al. [15]. Subsequent studies, predominantly focusing on manual or device-driven applications [2,4,11,15,16], have explored IAPV in neuromuscular disorders, which often lead to symptomatic chronic hypercapnia due to diaphragmatic and auxiliary muscle weakness. Research, including case reports and interventional studies, has indicated that IAPV is able to increase tidal volumes by up to 100%, effectively reducing hypercapnia [2,11], which can be supported by the actual results.

However, improved decarboxylation came at the expense of a temporary decline in oxygenation in all patients with CARDS, a trade-off that appears less problematic in predominantly hypercapnic lung failure but has to be taken into account. On the other hand, there were signs of amelioration of oxygenation during a longer period after IAPV and ERCC. Post hoc blood gas analysis during the following hours, when analyzing the IQR range, was indicating a slight rise in $PaO_2/FiO_2$ (S2 Table) and a reduction in $p_aCO_2$ levels. An explanation for this observed effect could be attributed to lung recruitment facilitated through the implementation of IAPV and ERCC. Bioimpedance measurements in one patient suggested increased ventilation in areas previously affected by atelectasis and pulmonary infiltration, suggesting potential lung recruitment and improved ventilatory distribution (S4 Fig). Improved long term oxygenation and decarboxylation by IAPV has been shown in other studies [17].

This study has several important limitations. It was designed as a single-arm, non-randomized pilot study, with a small sample size and a short observation period, which limit the generalizability of the results. It primarily focused on short-term effects and feasibility, leaving long-term outcomes unexplored. The inclusion criteria were specific to patients with hypercapnic lung failure predominantly in CARDS, which may restrict applicability to other respiratory conditions. Additionally, while the techniques improved decarboxylation, a temporary decline in oxygenation was observed. Manual application of the techniques could be variable across different operators.

It was designed as a single-arm, non-randomized pilot study, with a small sample size and a short observation period. The lack of a control group restricts any causal inferences. While the statistical analysis revealed relevant effects on tidal volumes and partial pressure of $CO_2$, future studies should include larger, controlled patient cohorts and evaluate the impact of these techniques on long-term outcomes, such as weaning success or ICU length of stay.

Of note, in some patients, the tidal volumes were elevated above the lung-protective range of 8 mL/kg predicted bodyweight by IAPV or ERCC. In such cases, an adjustment of the applied pressures would be necessary. However, that increase was not generated by inspiratory pressure opposing direct bronchio-alveolar barotrauma. Whether the externally applied pressure by IAPV or ERCC is harmful, if the range of 8 mL/kg PBW was exceeded, remains unknown.

In the actual study, patient No. 9 suffered from small bowel ischemia at day 9 post-study, caused by a torsion. We have no evidence, that the torsion was related to the study procedures as more likely ischemia would probably have emerged immediately after the intervention and not that late at day 9. Nonetheless, we must acknowledge that definitively excluding any association with the study cannot be assured with absolute certainty.

Finally, the actual project also involved the development of a prototype device for automating IAPV and ERCC explicit for intensive care and ARDS treatment, which has not been done before to our knowledge. This device, tested under laboratory conditions, demonstrated the ability to replicate the effectiveness of manual application.

It should be emphasized that the automated device was tested exclusively in a simulated ICU environment, which limits the applicability of these findings to real-life clinical settings.

The potential for those well-known physiotherapeutic, but technological poorly automated techniques is high, especially in resource-limited settings like intensive care units. Automation of these techniques not only promises to standardize and optimize their application but also potentially extends benefits in a wider range of clinical settings. The device's design and functionality are based on current technological capabilities and tailored to meet unique demands of critical care environments. Furthermore, we suggest that it would be technically feasible to implement a feedback mechanism that adjusts the applied pressures in response to tidal volumes or end-tidal CO2 levels, enabling patient-tailored respiratory support.

The amalgamation of clinical research and technological innovation shows potential for IAPV and ERCC to be used in various hypercapnic conditions, including weaning failure and exacerbated obstructive respiratory disease [10] and show a potential for IAPV and ERCC as rescue therapies if conventional ventilation and improvement strategies fall short [2,10].

## Conclusion

This exploratory study suggests that both IAPV and ERCC may be feasible and rapidly applicable techniques to enhance ventilation and reduce hypercapnia in patients with hypercapnic respiratory failure. In our limited sample, both manual and automated approaches were associated with increased tidal volumes. The automated prototype device, synchronized with the ventilator, demonstrated comparable effects to manual application under simulated conditions, indicating potential for standardized and automated use pending further clinical validation. Moreover, ventilator-synchronized compression may allow future adaptation to individual respiratory patterns, supporting the concept of personalized assistance strategies.

However, given the pilot nature of this study and the preclinical character of the automated prototype testing, further clinical studies are required to evaluate the efficacy, tolerability, and long-term safety of these interventions in broader patient populations.

## Supporting information

**S1 Fig. Flowchart on data collection and execution of the study.** This figure illustrates the process of data collection and the execution timeline and methodology of the study.
(PDF)

**S2 Fig. Air-filled cushion used for manual pressure measurement.** Self-constructed manual pressure sensor used during IAPV (one sensor) or ERCC (two sensors).
(PDF)

**S3 Fig. Visualization of pulmonary ventilation by PulmoVista.** This figure presents images of lung ventilation patterns as recorded by the PulmoVista system.
(PDF)

**S1 Table. Tidal volumes of individual patients before and during IAPV/ERCC, PEEP and Driving pressure values.** This table details individual patient tidal volumes and pressure settings pre- and during intervention.
(DOCX)

**S2 Table. Further outcomes.** This table summarizes secondary outcomes recorded during the study.
(DOCX)

**S3 Table. Tidal volumes without and during automated compression (ERCC, IAPV).** This table compares tidal volume data under baseline and automated compression conditions (laboratory simulations on a dummy).
(DOCX)

**S1 File. Die AktivEx Studie – Aktive manuelle Unterstützung der Exspiration bei invasiv beatmeten Patient*innen.**
(DOCX)

**S2 File. The AktivEx Study – Active Manual Support of Expiration in Invasively Ventilated Patients**
(DOCX)

**S3 File. TREND Statement Checklist.**
(PDF)

## Acknowledgments

The authors acknowledge the help of Prof. Monika Fuchs (HTW Berlin) and Lisa J. Weißmann (HTW Berlin), experts in textile engineering, who designed and manufactured the textile version of the prototype belt and the first air-filled cushions.

We extend our profound gratitude to NAW Berlin for their generous provision of the ventilation dummy and test lungs.

## Author contributions

**Conceptualization:** Denis Witham, Julius Valentin Kunz, Ann-Christin Krebs, Roland Körner, Paul Uwe Thamsen, Philipp Enghard, Julija Peter.

**Data curation:** Denis Witham, Julius Valentin Kunz, Ann-Christin Krebs, Philipp Enghard, Julija Peter.

**Formal analysis:** Denis Witham, Julius Valentin Kunz, Ann-Christin Krebs, Mareen Pigorsch, Philipp Enghard, Julija Peter.

**Investigation:** Denis Witham, Julius Valentin Kunz, Ann-Christin Krebs, Karoline Brückner, Roland Körner, Paul Uwe Thamsen, Philipp Enghard, Julija Peter.

**Methodology:** Denis Witham, Julius Valentin Kunz, Ann-Christin Krebs, Karoline Brückner, Roland Körner, Paul Uwe Thamsen, Philipp Enghard, Julija Peter.

**Project administration:** Denis Witham, Julius Valentin Kunz, Ann-Christin Krebs, Roland Körner, Philipp Enghard, Julija Peter.

**Resources:** Denis Witham, Roland Körner, Kai-Uwe Eckardt, Paul Uwe Thamsen, Philipp Enghard, Julija Peter.

**Software:** Ann-Christin Krebs.

**Supervision:** Denis Witham, Kai-Uwe Eckardt, Philipp Enghard, Julija Peter.

**Validation:** Denis Witham, Julius Valentin Kunz, Ann-Christin Krebs, Julija Peter.

**Visualization:** Julius Valentin Kunz, Ann-Christin Krebs, Mareen Pigorsch, Julija Peter.

**Writing – original draft:** Denis Witham, Julius Valentin Kunz, Ann-Christin Krebs, Julija Peter.

**Writing – review & editing:** Denis Witham, Julius Valentin Kunz, Ann-Christin Krebs, Karoline Brückner, Roland Körner, Kai-Uwe Eckardt, Paul Uwe Thamsen, Philipp Enghard, Julija Peter.

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
