## [Decision Letter · Decision Letter 0]

20 May 2025

PONE-D-25-02100Active Expiration Reduces Hypercapnia in Lung Failure – Results of the Prospective Interventional ActiveEx Study and Development of a Prototype Device for Automated ApplicationPLOS ONE

Dear Dr. Kunz,

Thank you for submitting your manuscript to PLOS ONE. After careful consideration, we feel that it has merit but does not fully meet PLOS ONE’s publication criteria as it currently stands. Therefore, we invite you to submit a revised version of the manuscript that addresses the points raised during the review process.

We look forward to receiving your revised manuscript.

Kind regards,

Yusuke Hoshino

Academic Editor

PLOS ONE

Journal Requirements:

2. Please upload a new copy of Figure 5 as the detail is not clear. Please follow the link for more information: "https://blogs.plos.org/plos/2019/06/looking-good-tips-for-creating-your-plos-figures-graphics/" https://blogs.plos.org/plos/2019/06/looking-good-tips-for-creating-your-plos-figures-graphics/

3. Please include captions for your Supporting Information files at the end of your manuscript, and update any in-text citations to match accordingly. Please see our Supporting Information guidelines for more information: http://journals.plos.org/plosone/s/supporting-information .

Additional Editor Comments:

There are quite a few clarifications suggested. Be sure to reply point-by-point in your revised submission.

Reviewers' comments:

Reviewer's Responses to Questions

**Comments to the Author**

1. Is the manuscript technically sound, and do the data support the conclusions?

Reviewer #1: No

2. Has the statistical analysis been performed appropriately and rigorously? 

Reviewer #1: No

3. Have the authors made all data underlying the findings in their manuscript fully available?

Reviewer #1: Yes

4. Is the manuscript presented in an intelligible fashion and written in standard English?

Reviewer #1: Yes

5. Review Comments to the Author

Reviewer #1: The authors present a small study to investigate Intermittent Abdominal Pressure Ventilation (IAPV) and Expiratory Rib Cage Compression (ERCC) in adult patients with hypercapnia. The paper is clearly written, but I have several concerns.

• The manuscript and the study protocol indicate that because this was deemed a pilot study, no sample size calculations were conducted. However, the protocol calls for 20 patients. However only 10 were enrolled, one withdrew, and nine were analyzed. There is no explanation for stopping enrollment after 10 patients.

• All patients were treated first with IAPV for twenty minutes, rested for 20 minutes, and then treated with ERCC. It would have been preferable to randomize the order in which the patients received the treatments.

• Reference number 12 lists the year incorrectly; it is The R Journal, 2023; 15(1):142-158.

• The most serious concern is that the statistical test used to compare the pre- and post- respiratory and hemodynamic outcome measures, the Brunner-Munzel test, is inappropriate for these data. This test is for data arising from two independent samples, but the data are collected before and after treatment on the same nine individuals, and thus are paired data, not from two independent samples.

6. PLOS authors have the option to publish the peer review history of their article (what does this mean? ). If published, this will include your full peer review and any attached files.

**Do you want your identity to be public for this peer review?** For information about this choice, including consent withdrawal, please see our Privacy Policy .

Reviewer #1: No

---

## [Author Response · Author response to Decision Letter 1]

11 Jul 2025

Dear Dr. Chenette,

Dear Academic Editors and Reviewers,

We are grateful for the opportunity to revise and resubmit our manuscript to PLOS ONE. We sincerely thank the academic editor and reviewer for their insightful comments and helpful suggestions. We have carefully addressed each of the points raised and made appropriate changes to the manuscript as required. All changes made to the manuscript have been marked in blue font for clarity. In the attached response to reviewers document, each point is addressed in detail.

We firmly believe that the insightful comments we received have substantially augmented the quality of our manuscript. Once again, we extend our deepest thanks for your expert guidance and unwavering support.

Sincerely,

The Authors

---

## [Decision Letter · Decision Letter 1]

30 Jul 2025

PONE-D-25-02100R1Active Expiration Reduces Hypercapnia in Lung Failure – Results of the Prospective Interventional ActiveEx Study and Development of a Prototype Device for Automated Application

PLOS ONE

Dear Dr. Julius Valentin Kunz,

Thank you for submitting your manuscript to PLOS ONE. After careful consideration, we feel that it has merit but does not fully meet PLOS ONE’s publication criteria as it currently stands. Therefore, we invite you to submit a revised version of the manuscript that addresses the points raised during the review process.

The reviewers believe there are some minor concerns which you should address before this manuscript is forwarded to the publisher. The suggestions and comments can be found at the end of this email.

We look forward to receiving your revised manuscript.

Kind regards,

Yusuke Hoshino

Academic Editor

PLOS ONE

Journal Requirements:

Additional Editor Comments:

Be sure to reply point-by-point in your revised submission.

Reviewers' comments:

Reviewer's Responses to Questions

**Comments to the Author**

1. If the authors have adequately addressed your comments raised in a previous round of review and you feel that this manuscript is now acceptable for publication, you may indicate that here to bypass the “Comments to the Author” section, enter your conflict of interest statement in the “Confidential to Editor” section, and submit your "Accept" recommendation.

Reviewer #1: (No Response)

2. Is the manuscript technically sound, and do the data support the conclusions?

Reviewer #1: Yes

3. Has the statistical analysis been performed appropriately and rigorously? 

Reviewer #1: Yes

4. Have the authors made all data underlying the findings in their manuscript fully available?

Reviewer #1: Yes

5. Is the manuscript presented in an intelligible fashion and written in standard English?

Reviewer #1: Yes

6. Review Comments to the Author

Reviewer #1: This revised manuscript is considerably improved. I just have a few minor comments regarding the revision.

line 150 - denoting relative effect by 'p' is confusing vis a vis p-values. please choose another symbol for relative effect.

Table 1: adjust column width so that the word 'ventilation' in the column heading is not split over two lines

line 205 - most measures are reported as median and IQR, but the manual compression pressure is reported as mean and

IQR. Why switch from median to mean for this variable? Also, reporting IQR along with a mean is unusual.

Generally one reports the standard deviation with means and IQR with medians.

Typos - next to last sentence in Results section of Abstract 'rapid' should be 'rapidly'

- line 137 - need a space between 'pressure' and 'was'

- line237 'from' should be 'for'

There are a few places in the revised manuscript where the marked changes remain (ie, were not accepted or cleared, such as in line 248)

7. PLOS authors have the option to publish the peer review history of their article (what does this mean? ). If published, this will include your full peer review and any attached files.

**Do you want your identity to be public for this peer review?** For information about this choice, including consent withdrawal, please see our Privacy Policy .

Reviewer #1: No

---

## [Author Response · Author response to Decision Letter 2]

26 Aug 2025

Response to Reviewer

We sincerely thank the reviewer for the careful re-evaluation of our revised manuscript and for the constructive minor comments. We very much appreciate the reviewer’s acknowledgment that the revised version has considerably improved, and we are grateful for the additional suggestions, which helped us to further enhance the clarity, precision, and overall quality of the manuscript. All comments have been carefully considered, and the requested corrections have been implemented accordingly. Our detailed responses are provided below, point by point:

R.1 Line 150 – Symbol for relative effect

We agree that the previous notation “p” for relative effect was confusing given its association with p-values. To avoid ambiguity, we have removed the symbol altogether. The results are now consistently reported as “relative effect” without using any additional symbol. We believe this wording is clearer and prevents potential misinterpretation by the reader.

R.2 Table 1 – Column width

The column width has been adjusted so that the term “ventilation” in the column heading is displayed on a single line.

R.3 Line 205 – Reporting of manual compression pressure

We thank the reviewer for pointing this out. The indication of “mean” was a typographical error. Manual compression pressure was in fact analyzed and reported as median (IQR), consistent with the other variables. This has now been corrected in the text.

R.4 Typos

Abstract, Results section: corrected “rapid” → “rapidly”

Line 137: inserted missing space between “pressure” and “was”

Line 237: corrected “from” → “for”

All previously marked changes have now been accepted and cleared.

Off-topic correction – Age data in Table

To further ensure patient anonymity, the exact ages originally shown in Table 1 have been replaced by decade ranges (e.g., “43” → “40-50”). This modification has no impact on the results or interpretation but strengthens data protection.

Changes to the Manuscript

Line 109: To ensure patient confidentiality, exact ages were not reported; instead, all ages were categorized into decade ranges (e.g., 43 years → 40–50 years).

We are grateful for these helpful suggestions, which have further improved the clarity and consistency of our manuscript.

Sincerely,

The Authors

---

## [Decision Letter · Decision Letter 2]

16 Sep 2025

Active Expiration Reduces Hypercapnia in Lung Failure – Results of the Prospective Interventional ActiveEx Study and Development of a Prototype Device for Automated Application

PONE-D-25-02100R2

Dear Dr. Kunz,

We’re pleased to inform you that your manuscript has been judged scientifically suitable for publication and will be formally accepted for publication once it meets all outstanding technical requirements.

Kind regards,

Yusuke Hoshino

Academic Editor

PLOS ONE

Additional Editor Comments (optional):

Reviewer #1:

Reviewers' comments:

Reviewer's Responses to Questions

**Comments to the Author**

1. If the authors have adequately addressed your comments raised in a previous round of review and you feel that this manuscript is now acceptable for publication, you may indicate that here to bypass the “Comments to the Author” section, enter your conflict of interest statement in the “Confidential to Editor” section, and submit your "Accept" recommendation.

Reviewer #1: All comments have been addressed

2. Is the manuscript technically sound, and do the data support the conclusions?

Reviewer #1: (No Response)

3. Has the statistical analysis been performed appropriately and rigorously? 

Reviewer #1: (No Response)

4. Have the authors made all data underlying the findings in their manuscript fully available?

Reviewer #1: (No Response)

5. Is the manuscript presented in an intelligible fashion and written in standard English?

Reviewer #1: (No Response)

6. Review Comments to the Author

Reviewer #1: Previous comments have been addressed

7. PLOS authors have the option to publish the peer review history of their article (what does this mean? ). If published, this will include your full peer review and any attached files.

**Do you want your identity to be public for this peer review?** For information about this choice, including consent withdrawal, please see our Privacy Policy .

Reviewer #1: No

---

## [Editor Report · Acceptance letter]

PONE-D-25-02100R2

PLOS ONE

Dear Dr. Kunz,

I'm pleased to inform you that your manuscript has been deemed suitable for publication in PLOS ONE. Congratulations! Your manuscript is now being handed over to our production team.

Kind regards,

on behalf of

Dr. Yusuke Hoshino

Academic Editor

PLOS ONE